# Development of Alternative Protein Sources in Terms of a Sustainable System

Paweł Sobczak [1,*], Józef Grochowicz [1], Patrycja Łusiak [1,*] and Wioletta Żukiewicz-Sobczak [2]

[1] Department of Food Engineering and Machines, University of Life Sciences in Lublin, 28 Głęboka Str., 20-612 Lublin, Poland; jozef@jozefgrochowicz.com

[2] Department of Nutrition and Food, Calisia University, 4 Nowy Świat Str., 62-800 Kalisz, Poland; wiola.zukiewiczsobczak@gmail.com

\* Correspondence: pawel.sobczak@up.lublin.pl (P.S.); patrycja.lusiak@up.lublin.pl (P.Ł.)

**Abstract:** Epidemiological studies of the population, changing dietary trends and climate change are the main factors influencing consumer choices. Although food overproduction and overconsumption are observed in the world, the proper nutrition of the population poses a problem. Despite satisfying bodily needs in terms of energy requirements, it is becoming increasingly difficult to balance diets with essential ingredients, such as protein. Traditional sources of proteins, due to changing dietary trends, are no longer attractive to consumers. Hence, global research is shifting towards alternative sources of protein. Therefore, this study aims to identify alternative sources of food protein from the perspective of the transformation of the food market. Scientific research, using innovative technologies, is targeting the previously underestimated sources of alternative raw materials and products, whose biological activity often astonishes the researchers themselves.

**Keywords:** alternative protein sources; health; sustainable development

## 1. Introduction

For several decades now, the market of developed countries has witnessed the phenomenon of overconsumption and westernization of the diet. Global food production and access to new markets allow consumers to choose from a wide range of products. For years, however, price has invariably been the decisive determinant of consumers' choices. This makes food producers reach for cheaper raw materials and technological methods that help reduce the unit cost of the product. Unfortunately, this trend only meets the economic needs of producers and consumers while significantly reducing the nutritional and health value of the product. Food becomes highly processed, satisfying the consumer's bodily needs only in terms of energy requirements. It did not take long to see the effects on global health, as evidenced by population epidemiological studies showing a high increase in diet-related diseases often referred to as civilisation diseases. Processed foods are rich in simple carbohydrates and saturated fatty acids, while low in essential dietary ingredients such as protein, fibre, vitamins and minerals. The climate change happening over the years has been another important issue indirectly limiting protein consumption. The spectre of environmental catastrophe is influencing the recommendations and advice of researchers. In addition, modern nutrition trends that exclude animal protein from diets are increasingly recommended by legislators.

The use of protein crops in the production of protein concentrates or isolates is frequently the subject of discussions in the European Union. Farmers' initiatives to find plant-based alternative protein sources are supported in many countries. One such source is soybean. Certainly, protein may come not only from animal feedstocks or plant sources that have been known for years but also from many other alternative sources that have not yet been used. Muscle tissue, which is rich in complete proteins with beneficial amino acid ratios for the human body, is the most popular source of protein. However, animal

husbandry in terms of sustainability is very expensive. Additionally, it produces a large amount of greenhouse gases, which adversely affects the environment, as evidenced by the ongoing discussions on climate change. The progressing climate change, which is evidently impossible to reverse, has been on the agenda of the European Commission, the WHO and the UN for many years. The recommendations related to counteracting food wastage in the context of hunger in developing countries and regional dietary profiling were published in the EAT-Lancet report, which already recommended healthy diets from sustainable food systems [1] at the end of 2019. The overconsumption of animal protein is also a core issue, as it leads to the overexploitation of the environment, on the one hand, and to the exacerbation of diet-related diseases, on the other. The modern approach to the presented civilisation problems is reflected in consumer choices, which have been increasingly conscious and reasonably justified. The partial or total elimination of animal protein, as well as the search for essential foods, is creating a new nutritional profile across the entire consumer population. Hence, alternative sources of protein can contribute to a reduction in meat production and fit into the concept of a sustainable economy.

Matters related to the search for alternative sources of protein have been increasingly present worldwide. In order to test the relevance of this phenomenon, publicly available databases were selected. The criteria for the acceptance of a database in the study included general availability, international coverage, the English language of the entries and the scientific nature of the database. Exclusion criteria were, in turn, a local or regional scope of the databases, the language of entries being one other than English, and the non-scientific nature of the databases. The following databases were selected: ScienceDirect, Elsevier, PubMed, Springer, and Frontiers.

This study utilises the method of assessment of the prevalence of the use of the phrase "alternative protein sources" in selected databases. The following databases were researched: ScienceDirect provided 600,374 results, Elsevier 15,333 results, PubMed 12,152 results, Springer 339,633 results, and Frontiers 26,765 articles. A visualisation of the results obtained is shown in Figure 1.

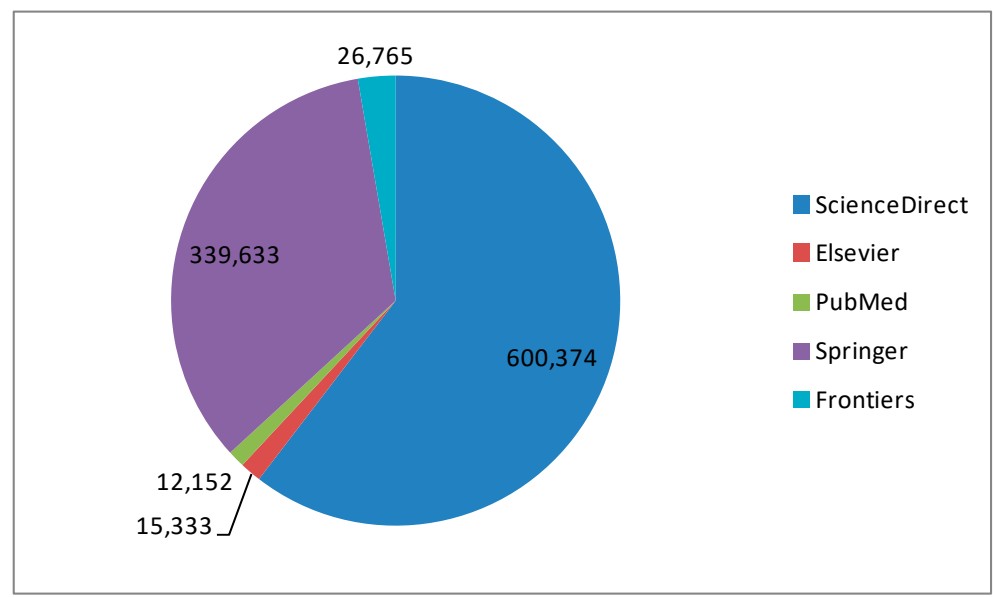

**Figure 1.** Frequency of the phrase 'alternative protein sources' in selected scientific databases [own elaboration].

As seen in the data presented, the topic of alternative protein sources has become increasingly prevalent in scientific databases worldwide. A great deal of interest in this issue has been observed in scientific databases.

The search for alternative sources of protein is also linked to an interest in its known forms, which are available locally or regionally in many parts of the world. For this reason,

foods and products used in Asian cuisine deserve attention, as they have been recognised as rich sources of protein for many years. Currently, there is a trend to reach for protein sources that were previously unappreciated by non-Asian consumers due to dietary culture and regional dietary profiling of individual populations around the world. Protein from insects, particularly popular in Asian markets, has been of great interest in recent years. Technological development and, in particular, bio-technological advancement have also made it possible to steer the vegetative processes of micro-organisms in a targeted manner. This is now used in many processes and production sectors worldwide. In this respect, the pharmaceutical industry is leading the way, although the issue of functional foods, dietary supplements and, most recently, novel foods seems to link pharmaceutics and nutrition to a significant extent. Table 1 shows the amino acid composition of other potential alternative protein sources that may limit animal protein production and the percentage reference to standard hen's egg protein.

**Table 1.** Amino acid profile of proteins derived from alternative sources of protein (g/100 g) with the percentage reference to standard hen's egg protein (%) [2–10].

| Protein Sources | Algae | Bacteria | Yeast | Fungi | Krill | Insects | Hen Egg Protein Standard Protein | FAO Scoring Pattern |
|---|---|---|---|---|---|---|---|---|
| Isoleucine | 4.7 (79.6%) | 3.3 (56%) | 2.5 (42.3%) | 1.8 (30.5%) | 2.5 (42.3%) | 3.8 (64.4%) | 5.90 | 2.8 |
| Leucine | 8.6 (102%) | 5.4 (64.2%) | 3.6 (42.8) | 2.9 (34.4%) | 4 (47.5%) | 6.5 (77.2%) | 8.41 | 6.6 |
| Valine | 6.2 (85.5%) | 4.2 (57.9%) | 2.7 (37.2%) | 2.2 (30.3%) | 2.6 (35.8%) | 5.2 (71.7%) | 7.25 | 3.5 |
| Lysine | 6.3 (105.8%) | 4.3 (72.2%) | 3.5 (58.8%) | 3 (50.4%) | 4.4 (73.9%) | 5.1 (85.7%) | 5.95 | 5.8 |
| Phenylalanine + Tyrosine | 9 (90.2%) | 5.8 (58.1%) | 4.1 (41.1%) | 3.1 (31%) | 5 (50.1%) | 9.7 (97.2%) | 9.97 | 6.3 |
| Methionine + Cysteine | 3.1 (50.8%) | 2.2 (36%) | 1.5 (24.5%) | 1 (16.4%) | 2.4 (38.9%) | 3.5 (56.8%) | 6.16 | 2.5 |
| Tryptophan | 0.9 (60.8%) | 0.8 (54%) | 0.6 (40.5%) | 0.3 (20.2%) | 0.7 (47.2%) | 1.2 (81%) | 1.48 | 1.1 |
| Threonine | 5.4 (125.5%) | 3.3 (76.7%) | 2.5 (58.1%) | 2 (46.5%) | 2.2 (51.1%) | 3.7 (86%) | 4.30 | 3.4 |

Own elaboration based on [2–10].

Considering changing dietary trends dictated by recommendations made in the context of climate change, as well as the epidemiology of diet-related diseases, it is necessary to look for new sources of protein in food. Therefore, this study aims to present and characterise selected alternative sources of protein in food.

## 2. Role of Protein in the Body

Protein is one of the components of a properly balanced human diet. Proteins are organic compounds of extremely complex structure that are essential for survival. They consist mainly of carbon 50–55%, oxygen 21–24%, nitrogen 15–18%, and hydrogen 6.5–7.3%. On a macro scale, in addition to the above-mentioned components, sulphur, phosphorus, calcium, iron, copper, iodine, zinc, and magnesium are also found in proteins. At the molecular level, the basic elements react to form amino acids, which are the elementary components of protein [9]. The biological role of protein is very important due to the number of functions it performs in the body. They are the building material for the construction of new and reconstruction of worn-out tissues, as well as a component of body fluids, such as blood and interstitial fluids. Moreover, they are components of digestive and tissue enzymes, as well as hormones. They are also a material for building immune bodies and a factor of acid–base balance in the body. According to many scientific sources, it is known that the demand for protein depends on age and the physiological state of the body, and a healthy adult with a normal body weight should consume 0.8–1 g of protein per 1 kg of body weight per day. Protein intake should account for 12–14% of the daily energy demand [10,11]. Protein is a specific molecule, which, unfortunately, the body is not able to accumulate or store. Therefore, this important ingredient should be

taken in accordance with developed standards and recommendations. The body's protein requirement increases by 1.5 to 2 times in children and adolescents, pregnant women, and lactating women [10,11].

Conventional food is currently considered not only a source of nutrients necessary to maintain the homoeostasis of the body but also a source of bioactive ingredients. It has been proven that food proteins are a source of bioactive peptides, which can regulate the physiological processes in the body, including all its systems, e.g., endocrine, immune, circulatory, nervous, and digestive. These findings are currently used, among others, in the production of functional food [12–14], and the peptides can be used in the prevention or even therapy of diet-related diseases [14,15]. Phosphopeptides derived from cow's milk's casein were the first biologically active peptides to be researched. In 1950, they were indicated by Mellander as factors, independent of vitamin D, conducive to the process of bone calcination in infants with rickets. In the 1970s, scientists began to consider food proteins not only as a source of amino acids necessary for the proper functioning of the body but also as precursors of biologically active peptides [16–18]. Bioactive peptides are fragments of amino acid sequences of source proteins that remain inactive in their precursors, while after the release from the parent proteins by proteolytic enzymes, they can act as modulators of a number of processes occurring in the body. A biopeptide should demonstrate a biological effect or effects that can be confirmed using appropriate measures and tests (e.g., pressure measurement), and in addition, these effects should be beneficial to health [19,20]. The biological activity of biopeptides constitutes the basic criterion for their division. They can affect the homeostasis of a human body and impact the reduction in blood pressure, exhibit antioxidant, antimicrobial, anti-amnesic, and opioid activity, influence the sensory properties of food, bind metal ions, and participate in their transport [9,12,17,21–28].

## 3. Innovative Solutions in the Production of Functional Food in the Context of Sustainable Development

Innovative solutions in the production of functional food are important considering the growing world population, which contributes to the continuous reduction in available nutrient-rich food sources, including animal protein, which is one of the most important nutrients in human nutrition. When looking for alternative sources of protein, Lemna minor (common duckweed) attracted the interest of the researchers. It is a water plant that was used in our country in the 1960s and 1970s, among others, to feed water poultry (ducks) and pigs [29,30]. Species of common duckweed can be found all over the world; they are cosmopolitan plants, adapting to different geographical areas and climatic zones. They can be found in all areas except for waterless deserts and areas subjected to permafrost. However, they grow most abundantly in tropical and temperate zones. Many species, though, are able to survive extreme temperatures. Lemna minor has been used in forage science for years as good fodder for swans and various species of Anseriformes. It is rich in protein, starch, vitamins and mineral salts. It has also been used to produce a concentrate, which, in turn, was used in nutritional tests on piglets. Lemna minor can be fed to farm animals, and it is also a good source of protein for aquaculture, and in the future, perhaps, for humans [15,31]. Currently, common duckweed has been used in biotechnology to obtain biologically active compounds, which are used in dietetics, phytotherapy and phytocosmetics. Common duckweed is also utilised in the biopharmaceutical industry [19,21,22].

The chemical composition of Lemna minor is not stable. In clean waters, where Lemna minor grows slowly, the biomass of the common duckweed contains relatively small amounts of nitrogen and phosphorus. Under conditions where nutrients are present in the water in abundance, Lemna minor grows intensively while at the same time absorbing large amounts of nutrients. Under such conditions, the content of nitrogen increases dramatically (3–5 times) in the biomass of the common duckweed, as do the content of other biogenic elements [18–22]. Lemna minor is characterised by a large variation in the content of protein, fat and dietary fibre, which results from the considerable impact of the environment and

the harvest phase, and, above all, the concentration of nutrients in the aquatic environment from which it is obtained. Common duckweed is characterised by a low dry matter content. This plant is able to obtain many macro- and microcomponents, such as Ca, Cl, K, Na, Si, N, H, C, Fe, Mg, Mn, Al, B, P, Cu, and Zn, from water, a property which can be used in the design of production conditions depending on the purpose and expected values of the final product in terms of mineral content [23,24]. Common duckweed also contains carotenoids. The aquatic habitat predisposes this plant to absorb various compounds found in water, and these might include profitable but also harmful ones such as, for instance, heavy metals. However, Lemna minor can be a very good supplement in food doses, as according to the FAO report [25] the concentration of such elements as cadmium, nitrogen, chromium, zinc, strontium, cobalt, iron, manganese, copper, lead, aluminium, and gold in this plant does not threaten animal and human health. The growth rate of the common duckweed, its chemical composition and the amount of the obtained biomass depend on many factors. These include the concentration of nutrients in the water, its temperature and pH, the amount of sunshine and the length of the day, as well as wind speed [25,32].

## 4. Protein Derived or Isolated from Arthropods and Parts Thereof

The meat production sector causes a number of adverse effects on the planet, including deforestation, soil erosion, threats to public health, loss of plant biodiversity and water pollution. Plant-based protein in the diet was supposed to be the answer to these destructive aspects of animal production; however, plant-based protein has been found to lack certain essential amino acids and to be less digestible than animal-based protein. Hence, the research in this area has been focused on insect proteins, which have been well-known for years, especially in Asian countries, but have not been appreciated in the West. It was found that this source can ensure sustainable protein production. The production of insects is sustainable as compared to animal production due to lower greenhouse gas and ammonia emissions released during the rearing processes. In addition, there is undoubtedly a lower demand for land and minimal use of water. These considerations are at the forefront of sustainable development and the prevention of climate change. Insects have a high protein content and excellent production efficiency as compared with other conventional proteins from traditional food groups. Therefore, according to the principles of the planetary diet, they fit perfectly into a sustainable diet and planet-friendly food production. The population growth and the increased demand for food have prompted researchers to explore alternative sources of protein suitable for human and animal consumption. The nutritional potential of alternative sources of protein is very important in this context, and so are more environmentally friendly methods of their production. The search for and consideration of the possibility of using alternative protein products in nutrition is also correlated with the assessment of raw materials and products currently of little interest among the European population. This might be exemplified by edible insects, which have been popular in Asian cuisine for centuries. They are used, among others, in the food, pharmaceutical and chemical industries. Insects can also be a part of the diet, as is the case in China, Japan, Mexico and South Africa. About 2000 species of insects consumed by humans are already known [32–34]. According to the FAO estimates, about 1.9 thousand insect species are consumed by an average of 2 billion people in more than 80 countries. Virtually all groups of insects such as crickets, termites, dragonflies, beetles, and caterpillars, in the form of adult individuals, pupae or eggs, are used for consumption purposes [6,35]. Based on data from the literature on the subject, crickets and locusts (Orthoptera), beetles (Coleoptera) and termites (Isoptera) were found to be the most commonly consumed edible insect species in the world [6,32,35].

Entomophagy, i.e., the practice of eating insects by humans, is common in some places, although it is traditionally rejected in others, mainly due to psychological barriers. Despite the fact that it is believed that insects were already consumed by prehistoric people (and thus fit into current food trends), in the EU, edible insects are considered to be novelty food and, therefore, applications and procedures should be followed in order to introduce them

to the market. Insects can also be used as fodder for livestock animals and as a support in increasing food production without placing additional pressure on the environment (indirect entomophagy). Until recently, insects were not perceived as part of a diet; hence, their consumption is limited to the unconscious consumption of products in which they are used as a food additive (e.g., cochineal) [36].

The market outlook for the middle of this decade indicates that the majority of demand will come from the feed sectors (i.e., pet food and the production of fodder for livestock). Undoubtedly, this sector is gaining momentum and its potential lies not only in food but also in fodder production, especially in the context of a circular economy [37].

From the nutritional point of view, insects are an important, though underestimated, alternative to nutrients supplied by conventional animal sources. They are characterised by being a source of energy, protein, and carbohydrates, having a high nutritional value and containing fat as well as vitamins and minerals. Numerous analyses of the chemical composition of insects have shown the variability between individual species, as well as, depending on the stage of their development, their habitat and type of food. The protein content in insects ranges from 5 to 77 g per 100 g; in many species of edible insects the protein constitutes more than 60% of the dry weight, and its highest content has been recorded in species from the order Orthoptera. Insect-derived protein has a digestibility comparable to that of a hen's egg white (77–98%) and is considered to be complete at a level comparable to milk and beef proteins. Insect proteins are a good source of the amino acids threonine, valine, histidine, phenylalanine, and tyrosine as well as tryptophan, and lysine. Edible insects are also a source of fat, the content of which varies from 10 to 50%. Orthoptera has an average fat content of 13%, Coleoptera (beetles) of 33%, and Rhynchophorus phoenicis Curculionidae larvae have a fat content of 67% of dry matter, which is higher than most conventional high-protein foods such as beef, poultry and eggs. When considering the composition of fatty acids present in insects, in terms of the degree of unsaturation, it is comparable to the composition of poultry and fish fat; however, insects are characterised by a higher content of polyene fatty acids [38–40]. In a study by Yang et al. [41,42], it was shown that the composition of fatty acids can be modelled via appropriate modifications in insect nutrition, and during the process of defatting insect meal, it is possible to obtain oil that can be used in human nutrition. Insects are characterised by a relatively low content of carbohydrates (0.1–5.3% of dry matter) and a very high content of fibre, mainly in the form of chitin. Additionally, the presence of vitamins and minerals adds to the high nutritional value of insects. When it comes to minerals, insects have the highest iron and zinc content. Copper, manganese, and magnesium are also present in smaller amounts, similarly to the small amounts of calcium. Insects are also a source of thiamine (0.1–4.0 mg per 100 g dry matter), riboflavin (0.1–8.9 mg), and cobalamin (0.5–8.7 µg per 100 g). They also contain folic acid, and in smaller amounts, retinol and β-carotene. Furthermore, insects are an indisputable source of peptides with antioxidant properties. In an experiment conducted by Zielińska et al. [43–45], the antioxidant activity of peptides obtained via in vitro digestion of edible insects belonging to five species was studied. The authors showed that the consumption of edible insects can be beneficial to health due to the strong antioxidant effect of the peptides obtained from them. The results revealed that the digested insects showed a higher antioxidant activity than other protein hydrolysates obtained from products of animal and vegetable origin [27–41,46]. Pathogenic bacteria found in insects are considered harmless to animals and humans due to genetic and physiological differences between host organisms. Zoonotic pathogens have also been found in feed for farmed insects; however, they do not pose a threat to the growth of production. While microflora harmful to livestock does not put insects at risk, it may develop intensively in their digestive tracts. Insects do not have specific prion diseases; although prions do not multiply in insect bodies, they can be transmitted at any stage of their development. For this reason, insects fed with fodder that does not contain products derived from human and ruminant tissues should not pose an epidemiological risk. As far as fungi and moulds are concerned, the most important factors in preventing their

occurrence are hygienic conditions of insect husbandry and fodder cleanliness. However, insects can accumulate heavy metals, dioxins, mycotoxins, and toxins in their bodies. The degree of heavy metal accumulation depends on the insect's species, its stage/phase of development, and the substrate used for feeding [38].

## 5. Biologically Active Peptides

Biologically active food-derived peptides are fragments of amino acid sequences that become active when released from precursor proteins. They are formed during digestion, fermentation or in vitro processes. A number of bioactive peptides were isolated from fish proteins, including peptides—inhibitors of the angiotensin-converting enzyme and antioxidant peptides, among others. Understanding the molecular aspects of the biological activity of peptides released from food proteins provides the basis for the use of these compounds as food ingredients or dietary supplements in the prevention of non-infectious civilization diseases. According to the WHO data, around 36 million people worldwide die each year from these diseases and an inappropriate diet is one of the main behavioural risk factors for the development of said diseases. Fish are mentioned among the raw materials and food products with a beneficial effect on the psychophysical condition of humans and are recommended for frequent consumption. Bioactive peptides can be released from protein precursors during hydrolysis with endogenous or exogenous enzymes. Biologically active peptides can also be chemically synthesised or obtained via genetic engineering methods. The source of bioactive peptides may consist of raw materials and food products, such as milk, fermented dairy drinks and cheeses, meat, eggs, plant raw materials and products, and raw materials from the seas. Biologically active peptides can, for example, exhibit inhibitory activity against many enzymes, including ACE, as well as antimicrobial, antioxidant, and opioid activity. They can also inhibit the platelet aggregation process, stimulate or inhibit the immune system response, bind metal ions and participate in their transport, as well as affect the sensory properties of food. Biologically active peptides derived from food proteins usually contain 2 to 20 amino acid residues. However, there are also those that may consist of a larger number of amino acid residues. Glycomacropeptide—a multifunctional peptide derived from κ-casein of milk—consists of 64 amino acid residues, the anti-cancer lunazine derived from soy is a 43-amino acid peptide, and 107–141-long amino acidfragment derived from haemoglobin is an antimicrobial peptide. Some of the bioactive peptides may exhibit their activity by acting locally in the gastrointestinal tract, e.g., as inhibitors of dipeptidyl peptidase IV, an enzyme that hydrolyses the incretin group hormones responsible for insulin response to a meal, or as agents conducive to maintaining intestinal mucosal integrity under oxidative stress, as in the case of lunazine. However, most biopeptides show their bioactivity only after crossing the intestinal barrier. Antioxidant H peptides have been identified in the hydrolysates of proteins from a number of fish, such as mackerel (*Scomber scombrus*), Maraud thistle (*Decapterus*), Alaska pollock (*Theragra chalcogramma*), tuna (*Thunnus*), salmon (*Salmo salar*), herring (*Aringus*), carp (*Cyprinus carpio*), tilapia (*Cichlidae*), silver carp, grass carp, fish of catfish order, yellowstripe scad, sardinella, fish of Johnius belangerii family, capelin, southern blue whiting, coho salmon, round scad, conger, yellowfin sole, flounder, Pacific hake, fish of Nemipterus hexodon genus, and Atlantic cod. Angiotensin-I-converting enzyme inhibitory peptides derived from fish proteins were first identified in 1986 in the hydrolysate of sardine muscle. Since then, ACE-I inhibitors have been identified in hydrolysates of fish proteins, from fish such as tuna, salmon, bonito fish, sardine, Alaska pollock, pink salmon, keta salmon, bigeye tuna, carp, shark, channel catfish, southern blue whiting, coho salmon, and katria [40–42]. Biologically active peptides are among the food components with high application potential in the treatment of civilizational diseases, such as hypertension. Bioinformatic analysis has shown that fish proteins are a potential source of peptides characterised by diverse biological activity. The ACE inhibiting ability, which plays a key function in reducing blood pressure, is the dominant activity occurring in all analysed protein sequences. The length of the protein precursor chain affects the number of bioactive peptides detected.

Among the studied fish proteins, the best potential source of bioactive peptides is actin-β of European perch, while the lowest concentration of biopeptides is found in the actin-β fragment of European whitefish. Knowledge of biologically active peptides derived from food proteins resulting from the latest scientific developments can be useful in food design, and it might significantly contribute to the programming of long-term preventive action, changing eating habits and preventing civilization diseases [40–44].

## 6. Protein Isolates and Concentrates from Leguminous Plants

Since the beginning of their existence, humans have been searching for ways to meet their needs. For this purpose, they have cultivated various plants, which constitute the first level of agricultural production, and, based on them, people have bred and reared animals, which, when eating plant products (sometimes also animal products), provide humans with a variety of products, mainly food ones. Vegetable protein is obviously cheaper than animal protein and, as such, is the basis of nutrition in poorer countries. According to Rutkowski et al. [30], most developed countries consume four times more meat and fats and six times more milk and eggs than developing countries. Vegetable foods (without fats) constitute only a third of all food consumed in developed countries. Worldwide, 80% of the protein produced is of plant origin. The inefficient conversion of vegetable protein into animal protein is reflected in the fact that only about 10% of the cereal produced is used directly in food production, while the remaining 90% is consumed in feed for the production of meat, milk, etc. It is assumed that the production of 1 kg of meat uses 4–7 kg of vegetable protein, while the production of 1 kg of milk protein and eggs uses 2.5–3.5 kg of vegetable protein. The conversion rate of protein feed into product protein is 30–40% in the case of milk, 15–25% for beef, about 20% for pork, while in the case of eggs and poultry, it is about 30%. The main source of vegetable protein, the basic food ingredient, consists of legume seeds cultivated worldwide [43,45]. It must be admitted though that products of animal origin are organoleptically close to the hedonic value optimum, while the culinary attractiveness of legume seeds is relatively low. Additionally, the consumption of animal products is often treated as a distinctive wealth indicator. As a result, in highly industrialised countries, the mainly consumed protein, usually even in excess, is that of animal origin. On the other hand, legumes, alongside cereals, dominate the diet of the inhabitants of the so-called developing countries, i.e., the poor ones. Advances in medicine, nutrition, and food science have made people aware of how much their health depends on what they eat. It is now generally recommended to increase the share of plant products in the diet while reducing the intake of animal protein and fats. The growing popularity of ecologically healthy food and vegetarian food is associated with an increased demand for high-protein raw materials, but the return of interest in legumes is mainly economically driven [43,45]. Leguminous plants, thanks to their nutritional and functional properties, are consumed in various forms. When dried, they are easy to store. They can be frozen and preserved. After initial preparation or processing into a puree, flour, flakes or protein preparations, they are of increasing use in food technology. They are components of mixtures of food concentrates, semi-processed meals or meals ready to eat after heating. They are increasingly used for the production of special-purpose foods, for example, for vegetarians, diabetics or people whose body is intolerant to gluten. In Poland, pea flour is produced in large quantities by the food concentrate industry. Worldwide, soybean has been, for many years, the most widely used among cultivated legumes. Research into the dietary use of the undervalued and forgotten lupine has been carried out successfully for several years in many countries in Europe, South America, Australia, Africa and Asia. In Chile and Peru, according to government schemes to increase the amount of protein consumed by the average citizen, lupine is added to bread. Vegetable protein preparations are increasingly widely used in food production. The raw material for their production can consist of legume seeds rich in fat (soybeans, peanuts, and lupines), as well as of those with a low fat content (peas, beans, field beans, etc.). Fairly neutral organoleptic properties of legumes are their additional advantage [43,44].

Protein preparations are most often produced in the form of concentrates and isolates using the so-called wet methods [44]. Special conditions are provided in the manufacture of protein concentrates to ensure that all constituents other than protein are extracted from the raw material. Protein isolates, on the other hand, are obtained via the acidic or alkaline extraction of the protein, which is then separated via precipitation or ultrafiltration. The technology used in the production of protein concentrates and isolates usually allows one to obtain products without the specific taste and smell of peas and unwanted antinutrients. Flour, grits, isolates and concentrates are increasingly used to produce textured protein products obtained via thermoforming or spinning. The first process is simpler and cheaper, although its products, in their structure, are not as similar to meat as the spun ones. They can serve as meat substitutes and meat-like products. New technologies for obtaining plant protein preparations make it possible to produce new types of preparations of programmed quality. Different protein content, its form and physicochemical properties have a varying impact on the quality of the final product. In order to obtain the best results, many studies have been devoted to the analysis of the functional properties of proteins. Protein preparations are most often treated as a component of the product or an additive, and the quality of the finished product depends on the quality characteristics of all components and their mutual interaction. Soybean protein preparations (and potentially of other legumes such as lupine) are mostly used in bakery and meat processing. Obviously, the presence of a substitute cannot reduce the consumer acceptance of the product; it should, however, reduce its price. The main purpose of introducing a protein preparation as an additive is to obtain a good product from a raw material of lower quality. Protein preparations can be used primarily in meat products such as steamed sausages, breakfast meats, offal meats, pates, burgers, meatballs, etc. They can also be added to culinary minced meat, raw smoked sausages, protein and fat emulsions, and as an additive improving the functional properties of mechanically deboned meat. They are used as ingredients in brining mixtures, as well as for marinates and sauces, while textured preparations are used as replacements and equivalents of meat. The use of vegetable protein preparations in the bakery industry already has a long history. Soy flour and grits are added to the dough in order to prolong the freshness of the baked goods, brighten the colour and improve the crumb structure, darken the skin colour, and increase the nutritional value by increasing the protein level. Soy preparations are added to pasta mainly to increase its nutritional value and improve the amino acid balance by enriching it with deficient lysine [43–45].

## 7. Fodder Legumes

In the past, lupine was used exclusively as cattle feed, but since 1955, the use of lupine has been patented for the production of different edible masses—cocoa and coffee substitutes. Currently, it is most often used as an addition to energy bars for athletes or as a substitute for soybean in products intended for vegetarians and vegans.

There are more than 200 species in the Lupinus L. genus. Narrow leaved lupine occurs natively in the Mediterranean. The first attempts to use it as feed date back to the late 19th century. After World War II, this species of lupine was domesticated and many modern varieties have been cultivated. Yellow lupine grows in Sicily in the Etna region and in the Peloponnese, and its occurrence is associated with acidic rock substrate. In the Middle Ages, it was cultivated as a decorative plant, and its cultivation gained more importance in Germany at the end of the 19th century. At that time, its cultivation also began in Greater Poland, where it was used as green fertiliser. It was not until the discovery of low alkaloid forms in 1927 that a new stage in the domestication of this species began. In Poland, even before World War II, the first low alkaloid yellow lupine variety, i.e., Bielański Pastewny, was obtained, and, in the 1970s, the first fusariosis-resistant variety was introduced into cultivation. Andean lupine has been cultivated on the Andean plateau for 3000 years, as well as in other areas of South America, where it gives unstable yields (1.0–4.0 t/ha). It is the most valuable species of lupine due to its high protein (40–50%) and fat (12–20%) content. Lupine has the highest protein content, among the plants grown in

Europe (Table 2). Its protein has a beneficial composition of exogenous amino acids, but the deficiency of sulphuric amino acids, especially methionine, limits its value.

Lupine and field bean are characterised by the lowest genetic and fluctuating protein content.

Field bean is a plant originating from Central Asia. As a cultivated plant, it was already known in Antiquity. The greatest concentration of its cultivation is found within the maritime climate range, as the plant requires a large amount of water for its growth and development. It is a plant with the highest yielding potential; it produces 4–5 tons of seeds per hectare. In the Mediterranean countries, faba bean seeds are used as food for people. These seeds contain 26–30% of protein of high biological value. Unfortunately, the seeds also contain numerous substances that limit the availability of nutrients, interfere with the functioning of glands and internal organs, impair health, and reduce the productivity of animals. Field bean seeds are not recommended for feeding dairy cows. In the feeding of calves over the age of 12 weeks and fattening, field bean seed meal may constitute up to 10% of concentrate mixes. The straw of the faba bean is stiff, which is why the animals are reluctant to eat it. The cultivation of faba bean for green matter in pure sowing makes little economic sense due to the low yield, especially in years with low rainfall, and the very high cost of seed. Field bean plays an important role in crop rotation, interrupting the frequent succession of cereals. In addition, on 1 ha, it can leave approx. 5–8 t of organic mass in the form of postharvest residues, with approx. 80 kg of nitrogen, 6 kg of phosphorus and 120 kg of potassium. The network of numerous tubules that remains after the deep root system dies down allows the soil to be aerated and to store a large amount of water. Plant breeding is dominated by the work carried out with the aim of increasing the fertility of varieties. Additionally, low tannin varieties can be currently found on the market, which perhaps could be used for the production of high protein food. [43–45,47,48].

**Table 2.** Content of protein and more important amino acids in leguminous seeds [49].

| Species | Protein | Limiting Amino—Acids (mg/g N) | | | | |
|---|---|---|---|---|---|---|
| *Vicia faba* L. Faba bean | 25.0 | Met | Cys | Thr | Try | Val |
| | | 25.0 | 50 | 210 | 0 | 280 |
| *Lupinus luteus* L. Yellow Lupin | 44.3 | 44.3 | 90 | 225 | 60 | 250 |

Given the above, plant-based meat alternatives (PBMAs), which are semifinished or finished foods (e.g., texturized chops and sausages, vegan pâtés), are of scientific interest. The raw material for the production of such substitutes consists of extruded protein from soybeans, peas, wheat or mould biomass (Fusarium venenatum, now known as Quorn) [48–52]. The range of meat substitutes available on the market is characterised by limited diversity (most products are soybean-based), texture and organoleptic characteristics that in many cases differ from the original, reduced nutritional value (high content of fat, carbohydrates, food additives such as methylcellulose), and a higher price than the meat equivalent.

## 8. Conclusions

Recent years have progressively demonstrated the primacy of nature over human expansion in its environment. The year 2021 showed a new reality in many areas of human life. The fight against the SARS-CoV-2 pandemic has clearly affected the global public health system. Unfortunately, the dilemmas that already existed and concerned researchers in previous years are still very much valid. Food production on a macro scale evidently has an impact on the environment, as reported by scientists in the field. For this reason, studies are carried out and reports are made that are then implemented in the form of suggestions aimed at mitigating, in accordance with the recommendations, the adverse

effects at the interface between health, food and nutrition, and the environment. Due to the ongoing climate changes, as well as worrying reports on the health of the population, a transformation of the food market is a necessity. Based on the increasingly frequent claims, it is known that plant-based meat alternatives are healthier for both humans and the planet. Food production is one of the main contributors to climate change and the destruction of ecosystems, as it is the source of high greenhouse gas emissions and freshwater consumption. It leads to deforestation, loss of biodiversity, and disruption of the phosphorus and nitrogen cycles in nature. However, these threats from agriculture do not spread evenly. Animal husbandry requires more resources per unit of food weight than crop production. This is due to the fact that the efficiency of animal protein production is extremely low. Farmed animals worldwide consume 4.6 Gt of elemental carbon per year, but their dairy and meat products contain only 0.12 Gt of carbon. That is only 2.6% of what the animals eat. This is because most nutrients are used for life processes and not for meat and milk production. For this reason, meat production is very inefficient. Globally, on average, 250 kg of protein will be obtained from 1 hectare of wheat, but only 10 kg of beef protein will be obtained from 1 hectare of pasture. Additionally, although about 80% of the agricultural land is intended for animal production, it yields less than 40% of the protein and 20% of the calories consumed by people [53]. In addition, animal production involves enormous emissions of pollutants and greenhouse gases. This inefficiency means that as the population develops and a growing number of people want to "eat" more meat, its production becomes an increasing problem for the environment. Not to mention the health effects of consuming excessive amounts of meat and animal fats. According to Farvid et al. (2021) [54], systematic dietary consumption of red meat can lead, among others, to an increased risk of chronic diseases such as diabetes or cancers, including breast, prostate, colorectal, and kidney cancer, as well as cardiovascular disease [44]. In addition, global animal production also raises ethical questions. Another consideration is the issue of health and food safety, which is linked to the issue of antibiotic resistance and the emergence and spread of drug resistant, pathogenic bacterial strains (e.g., Escherichia coli, Salmonella Enteritidis) [50–59].

Bearing in mind the above aspects, it is becoming extremely important to search for alternative sources of protein, especially taking into account its production in a sustainable manner. Such examples include insects and Lemna minor, the acquisition of which does not cause such a significant environmental degradation and contributes to the reduction in civilisation-related diseases.

The recommendations made in the previously cited report [1] concern the urgent transformation of the global food system. The main focus of the report is on the destructive impact of the current food production on the climate. At the same time, a diet that is beneficial to the globe is recommended, along with its principles and product groups in the so-called "planetary diet", among which is the use of alternative sources of protein. Global food production threatens climate stability and contributes to environmental catastrophe. Agriculture generates excessive amounts of greenhouse gases that contribute to global climate warming. At the same time, the world's freshwater resources are in steady decline. A planetary diet is not only a solution to negative climate impacts, but can also be very important in the prevention of diet-related diseases. The assumptions presented by the EAT platform relate to a model of sustainable consumption that prioritises plant-based production over animal-based one. In addition, the choice of local, seasonal products associated with the region's culture is recommended. According to experts, the planetary diet aims to reverse the effects of the current global food production system. The implementation of this dietary model worldwide could significantly reduce the proportion of overweight and obese people, reduce overconsumption in highly developed countries and contribute to the fight against hunger in poor countries. According to the EAT-Lancet experts, a planetary diet would not only help avoid serious environmental degradation but also prevent the death of some 11 million people annually [1]. Hence, the search for alternative sources of non-animal protein is strongly linked to the above assumptions.

The meat industry producing protein for human consumption has a significant negative impact on the environment through the use of large amounts of water (e.g., for irrigation of crops from which fodder is produced, watering animals, and cleaning them), energy (e.g., for lighting, heating, cooling, and storage), and the production of large amounts of greenhouse gases (carbon dioxide, and methane). In addition, the consumption of red meat has been linked to an increased risk of chronic diseases such as diabetes, as well as breast, prostate, colorectal and kidney cancer, and cardiovascular diseases [26,55]. Mass animal production also raises ethical and safety concerns relating to the emergence and spread of drug-resistant, pathogenic bacterial strains (e.g., Escherichia coli, Salmonella Enteritidis) [3,5,8,11,15–17,49–59]. Unfortunately, traditional forms of dietary protein are based on animal husbandry and, in particular, products derived from animals for slaughter. For this reason, vegetarian and vegan diets are becoming increasingly popular among people, and the reasons for switching to these diets fall into two main categories: moral and health-related. Moral reasons are associated with animal suffering, especially in the context of industrial animal breeding and slaughter, while others are motivated to switch to these diets due to world hunger—excessive meat production consumes vast plant resources, which limits the amount of food accessible to developing countries.

On the basis of the database analysis presented in this paper, it seems obvious that alternative forms of protein are phrases very frequently used in scientific and research databases, as well as in less professional ones. The use of alternative protein sources seems to be an important consideration in global nutrition. Hence, research related to this topic should be taken seriously.

Opinions persuading consumers to switch to a vegetarian or vegan diet are more and more frequently heard and they are often sceptically received by the population. According to the researchers, a better solution is to focus on plant-based meat equivalents. Providing access to such products and lowering their price can be truly effective. All the more so because small changes can bring about great benefits. In general, the production of plant food applies much lower pressure on the environment as compared to animal food production. Plant-based meat substitutes are not only produced more sustainably, resulting in lower greenhouse gas emissions, and smaller water and land use, but also bring a number of health benefits.

It should also be noted that there is a fairly wide range of plant-based protein products on the food market, but such products bear the hallmarks of highly processed foods. Therefore, where is the source of the problem? Perhaps the technological methods currently used are not properly designed for this type of raw materials. Therefore, all over the world, research is being conducted to create new methods of producing raw materials that can provide the population with alternative sources of protein in a planet-friendly manner. Functional food is of the utmost importance, both in providing adequate nutrients to humans and animals, protecting them from selected diseases and, perhaps, in the future, helping to solve the world's hunger problem. Still, the search for new sources and/or active ingredients remains an important challenge for the entire scientific community [16,27,60–64].

**Author Contributions:** Supervision, methodology, P.S.; resources, J.G.; writing and editing, writing—original draft preparation, P.Ł.; visualization, conceptualization, funding acquisition, W.Ż.-S. All authors have read and agreed to the published version of the manuscript.

**Funding:** This research received no external funding.

**Institutional Review Board Statement:** Not applicable.

**Informed Consent Statement:** Not applicable.

**Data Availability Statement:** Not applicable.

**Conflicts of Interest:** The authors declare no conflict of interest.

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
