# Peer review of "Development of Alternative Protein Sources in Terms of a Sustainable System"

_sustainability, doi:10.3390/su151612111_

Round 1

Reviewer 1 Report

1. Aim of the study can be included in introduction. Proteins and sustainability that is given in title is not explained in detail in the introduction

2. Duckweed importance on protein and its quality is not provided. why duckweed is suddenly there without laying importance on protein and also the references are old.

3. toxicology and safety aspects of insect proteins could have been covered. explain how is it sustainable

4. line 203 "foodderived" to be separate and line 282 "socalled" 

5. The sources selected like fish protein - how are they sustainable sources?

6. Line 269-276 can be explained using sustainability

7. Conclusion can also be focused on sustainability 

8. Many words are merged throughout document. Please rectify.

9. Recent references and materials can be cited

10. Overall grammar can be improved.

11. The aim and title does not match well, so manuscript can be revised accordingly.

Grammar can be rechecked for the manuscript. Long sentences can be reduced.

Author Response

Responds to the Reviewers:

We would like to thank the Reviewers for his interest in our work and for his helpful comments that helped us to greatly improve the quality of manuscript. We have tried to do our the best to respond to all the points raised. As indicated below, we have checked all the comments provided by the Reviewers and have made necessary changes according to them.

Reviewer 1

  1. Aim of the study can be included in introduction. Proteins and sustainability that is given in title is not explained in detail in the introduction

According to the suggestion we moved aim of the study to the introduction chapter, and added some information of protein production in sustainability system to the introduction chapter. All added information are market in to the text of manuscript.

  1. Duckweed importance on protein and its quality is not provided. why duckweed is suddenly there without laying importance on protein and also the references are old.

As stated in the title of the chapter "searching for new sources of protein", duckweed appeared as its source for this reason. The quality of this protein can vary, which depends on many factors and its growth intensification. These relationships are described in the text of the work together with the references.

  1. toxicology and safety aspects of insect proteins could have been covered. explain how is it sustainable

And this is very important for human health. We included this information in our study together with the references source. Such a mention can be found in Chapter 4.

  1. line 203 "foodderived" to be separate and line 282 "socalled" 

We corrected these remarks, and also sent the whole text for grammatical correction.

  1. The sources selected like fish protein - how are they sustainable sources?

In search of new sources of protein, alternative to animal production, we also choose fish protein in our considerations as a source of much simpler production in the sustainable development system in relation to animal production, which in this system is more difficult and cost-intensive. That is why fish protein appeared in the study.

  1. Line 269-276 can be explained using sustainability

We have included such information on purpose to show what is the current use of the vegetable protein produced. But improving and improving the production of plant protein and other alternative sources of protein in the future will certainly improve the sustainability aspect.

  1. Conclusion can also be focused on sustainability 

This part was corrected according reviewer comments.

  1. Many words are merged throughout document. Please rectify.

We corrected these remarks, and also sent the whole text for grammatical correction.

  1. Recent references and materials can be cited

Yes. We included 5 references from 2022 and 5 from 2021.

  1. Overall grammar can be improved.

We corrected these remarks, and also sent the whole text for grammatical correction.

  1. The aim and title does not match well, so manuscript can be revised accordingly.

We improved the manuscript according to the reviewers suggestions, so we hope it is more suitable to the title and to the aim of work.

Reviewer 2 Report

The manuscript content is not enough to publish. This is not a comprehensive article. It should be revised significantly before acceptance.

English should be improved and concised.

Author Response

Responds to the Reviewers:

We would like to thank the Reviewers for his interest in our work and for his helpful comments that helped us to greatly improve the quality of manuscript. We have tried to do our the best to respond to all the points raised. As indicated below, we have checked all the comments provided by the Reviewers and have made necessary changes according to them.

Reviewer 2

  1. Line 60 and 61: Please provide the references.

We added the references it to the text [2,3].

  1. Line 110: The chemical composition of Lemna is not stable, what is the reason for that?

Yes we improved this information. The text is market in the manuscript.

  1. Line 115: No reference has been given.

We added the references it to the text [16].

  1. Line 249: Abbreviation of ACE should be used as similar to line 221.

We changed the abbreviation in the line.

  1. Line 202: Why did you include 6. Biologically active peptides section?

Due to the fact that the article concerns alternative sources of protein in food, according to the authors, it is important to outline the basics related to the biological activity of this important ingredient in the human diet. Understanding the molecular aspects of the biolog-ical activity of peptides released from food proteins provides the basis for the use of these compounds as food ingredients or dietary supplements in the prevention of noninfectious civilization diseases.

  1. Line 422: Please provide the references.

We added the references it to the text [45].

  1. Comment 1. Section 8 fodder crops: There is a lot of unnecessary information in this section, please make it concise.

Due to the fact that legumes are one of the main sources of vegetable protein, the authors decided to present the origin of their appearance in the human diet. According to the authors, the information was presented very concisely. However, if some data are omitted in a foreign form, the text may lose its meaning.Comment 2. More sustainable sources of proteins should be included.

  1. Comment 3. There is only one Table in entire article. Please include more tables and figures.

We added one table and one figure more to the manuscript.

  1. Comment 4. This article is not comprehensive enough to be published. It needs to be revised significantly.

The article has been extended and corrected according to the reviewers and editors.

We improved the manuscript according to the reviewers suggestions, so we hope it is more suitable to the title and to the aim of work.

Round 2

Reviewer 1 Report

Comments

Line 48 – expensive spelling error

Line 54 – Expand EAT

Line 191 – Please describe how protein derived or isolated from, arthropods and parts thereof are sustainable ??

Comments:

Line 48 – expensive spelling error

Line 54 – Expand EAT

Line 191 – Please describe and justify how protein derived or isolated from, arthropods and parts thereof are sustainable ??

Please check grammar overall throughout the document

Author Response

We would like to thank the Reviewers for his interest in our work and for his helpful comments that helped us to greatly improve the quality of manuscript. We have tried to do our the best to respond to all the points raised. As indicated below, we have checked all the comments provided by the Reviewers and have made necessary changes according to them.

Line 48 – expensive spelling error

The document was grammatically checked and all mistakes were corrected

Line 54 – Expand EAT

the EAT-Lancet report recommendings healthy diets from sustainable food systems

Line 191 – Please describe how protein derived or isolated from, arthropods and parts thereof are sustainable ??

The text was added to the manuscript:

Meat production sector causes a number of adverse effects on the planet, including deforestation, soil erosion, threats to public health, loss of plant biodiversity and water pollution. Plant-based protein in the diet was supposed to be the answer to these de-structive aspects of animal production, however, plant-based protein has been found to lack certain essential amino acids and to be less digestible than animal-based protein. Hence, the research in this area has been focused on insect proteins, which have been well-known for years, especially in Asian countries, but have not been appreciated in the West. It was found that this source can ensure sustainable protein production. Production of insects is sustainable as compared to animal production due to lower greenhouse gas and ammonia emissions released during the rearing processes. In addition, there is undoubtedly a lower demand for land and minimal use of water. These considerations are at the forefront of sustainable development and the prevention of climate change. Insects have a high protein content and excellent production efficiency as compared with other conventional proteins from traditional food groups. Therefore, according to the principles of the planetary diet, they fit perfectly into a sustainable diet and planet-friendly food production.

Reviewer 2 Report

The manuscript could be accepted for publication.

English has been improved.

Author Response

(The authors gave the same response as above.)
